# Effects of essential tremor on longevity and mortality rates in families

**Onur Emre Onat**[1,2*], **Faruk Ustunel**[1,3], **Cem Akbostanci**[4], **Kivilcim E. Doganyigit**[5], **Merve Sen**[6], **Emre Can Gunaydin**[2], **Kaya Bilguvar**[7,8], **Muhittin Cenk Akbostanci**[9]

**1** Beykoz Institute of Life Sciences and Biotechnology, Bezmialem Vakıf University, İstanbul, Türkiye, **2** Department of Biotechnology, Institute of Health Sciences, Bezmialem Vakıf University, İstanbul, Türkiye, **3** Department of Drug Discovery and Development, Institute of Health Sciences, Bezmialem Vakıf University, Istanbul, Türkiye, **4** School of Psychology, Washington State University, Pullman, Washington, United States of America, **5** Department of Biomolecular Engineering, School of Engineering, University of California Santa Cruz, California, United States of America, **6** Centre for Ophthalmology, Institute for Ophthalmic Research, Universitatsklinikum Tübingen, Tübingen, Germany, **7** Department of Medical Genetics, School of Medicine, Acıbadem Mehmet Ali Aydınlar University, İstanbul, Türkiye, **8** Departments of Neurosurgery and Genetics, School of Medicine, Yale University, New Haven, Connecticut, United States of America, **9** Department of Neurology, Faculty of Medicine, Ankara University, Ankara, Türkiye

* onur.onat@bezmialem.edu.tr

## Abstract

Essential Tremor (ET) is a common movement disorder characterized by action tremors, primarily affecting the hands and head. lthough previous studies have suggested potential links between ET and aging-related diseases, its relationship with longevity remains unclear, with conflicting evidence in the literature. To investigate this association, we analyzed data from 1,493 individuals across 145 families, encompassing both ET-positive (ET+) and ET-negative (ET−) participants. Using comprehensive statistical methods, including survival function estimation and regression modeling, we examined the potential influence of ET on lifespan. The median age of our participants was 67 years (IQR 54–77). Among deceased individuals, those with ET had a higher median age at death (80 years, IQR 70–86) compared to their ET− counterparts (70 years, IQR 59–77). Living ET+ participants also demonstrated slightly higher median ages (63 years, IQR 53–74) than living ET− individuals (60 years, IQR 49–71). Survival analysis revealed a significantly prolonged lifespan for ET+ individuals compared to ET− individuals (log-rank p = 1.11 × 10$^{-23}$). Furthermore, hazard ratio (HR) calculations indicated a reduced risk of mortality for the ET+ group (HR = 0.44, CI95% = 0.37–0.52), particularly among males. These findings suggest that ET may be associated with increased longevity, though the underlying biological mechanisms remain unclear. Further research is essential to elucidate the processes contributing to this relationship and to explore its implications for understanding aging and neurodegenerative disorders.

## Introduction

Movement disorders encompass a wide range of neurological conditions characterized by abnormal body movements, which may be voluntary or involuntary [1]. Among these, essential tremor (ET) is one of the most common adult-onset movement disorders, primarily

---

**Data availability statement:** All relevant data are within the paper and its Supporting Information files. The raw data supporting the conclusions of this article can be requested from the authors. The R Markdown and Python codes used for the analysis are accessible on GitHub (https://github.com/farukustunel/et-longevity-analysis). These codes include scripts for data processing, statistical analysis, and visualization.

**Funding:** The author(s) received no specific funding for this work.

**Competing interests:** The authors declare that the research was conducted without any commercial or financial relationships that could be construed as a potential conflict of interest. This does not alter our adherence to PLOS ONE policies on sharing data and materials.

recognized by its characteristic 4–12 Hertz postural or kinetic tremor [2]. This tremor typically affects the hands and arms and often presents without other neurological conditions [1,3]. Although ET can emerge at any age, it predominantly appears later in life [1,4,5]. Globally, ET affects approximately 1.33% of the population, according to a meta-analysis aggregating data from 42 prevalence studies across 23 countries [2]. Its prevalence increases with age, rising to 5.9% among individuals aged 60–65 and reaching 11.4% in those 80 and older. Despite its prevalence, ET remains poorly understood in terms of its relationship with lifespan [2].

ET is a chronic and progressive condition, with tremor severity worsening over time and frequently accompanied by co-morbidities. These factors contribute to reduced functional capacity and heightened frailty [6]. Although the exact genetic mechanisms behind ET are elusive, both environmental factors and a strong hereditary component are implicated in its development [7]. Research highlights potential cerebellar involvement, with structural abnormalities such as dendritic swellings and Purkinje cell heterotopias observed in individuals with ET [8,9]. These findings suggest that cerebellar dysfunction may play a significant role in the disorder, although the precise neurobiological pathways remain unclear.

The clinical presentation of ET is highly heterogeneous in nature, showing variability in its etiology [10], age of onset [11], clinical features [12], and response to pharmacological treatment [13,14]. Historically considered as a mono-symptomatic disease, ET is now recognized as a spectrum of disorders sharing the common feature of kinetic tremor [15]. This expanded view of ET includes a wide range of motor symptoms (i.e., tremors in hand, head, voice, lower limbs, and other body parts and mild gait ataxia), non-motor symptoms (i.e., mild cognitive deficits and dementia), psychiatric disorders (i.e., anxiety, depression, and sleeping problems), and sensory nervous system symptoms (i.e., partial hearing loss) [3,4,16]. This broad spectrum of symptoms reflects the intricate and multifaceted nature of ET, making diagnosis and management challenging.

While neurodegenerative diseases are generally associated with shortened life expectancy, the impact of ET on longevity remains unclear [17–20]. Early observational studies such as that by Minor, suggested a potential survival advantage for individuals with ET [17,18], yet subsequent research has produced contradictory findings. For example, a longitudinal retrospective study conducted in 1984 by Rajput *et al.*, on 266 ET patients, found no evidence of increased longevity. However, this study was limited by its relatively young cohort and insufficient follow-up into advanced age [19]. Conversely, a more comprehensive study by Jankovic *et al.* in 1995 examined 201 ET patients and compared them to 465 age-matched controls in a longitudinal prospective design. Their results indicated significantly increased longevity among ET patients with a relative risk [RR] of 1.59, ($p$ = 0.01) [20]. More recently, a population-based prospective study conducted in three villages in central Spain in 2007 by Louis *et al.* suggested that ET may even increase mortality risk under certain conditions [21]. These conflicting results highlight the need for further investigation into the complex relationship between ET and lifespan.

To address these uncertainties further investigation into the relationship between ET and mortality is crucial. There is a clear need for more rigorous, family-based studies to investigate how ET might influence survival rates, particularly by accounting for genetic factors and controlling for confounding variables such as diet and socioeconomic status. Additionally, there is a lack of data exploring gender-specific differences in mortality and how such differences could contribute to our understanding of ET's impact on longevity. In our comprehensive family-based retrospective study, we aimed to address this gap by comparing mortality risk in ET patients to their family members. By comparing survival outcomes among individuals with and without ET, and accounting for potential confounding factors, this work seeks to provide

new insights into the potential link between ET and extended lifespan. In this study, we examined the pedigrees of 145 probands and tested the association of ET with longevity in 1,493 subjects. Our findings suggest that ET could indeed be associated with longer life expectancy, offering an intriguing avenue for further research into the underlying mechanisms that might confer this potential longevity advantage. These findings, if confirmed in large-scale longitudinal studies and biological investigations, could have significant implications for public health, particularly in understanding the complex interplay between ET, aging and survival.

## Materials and methods

### Study participants

This study is a part of an ongoing project "Genetics of essential tremor in Turkish families: Identification of the causal variants." The primary aim of the study is to explore the genetic underpinnings of essential tremor (ET) in familial cases across Türkiye. Written (signed) informed consent was obtained from all participants at the time of enrollment. All procedures involving human participants adhered to the ethical standards set by the institutional and national research committees, as well as the 1964 Helsinki Declaration and its later amendments, or comparable ethical guidelines. Ethical approval for the study was obtained from the Institutional Review Boards of both Bilkent University and Ankara University, under the regulations and guidelines issued by the Turkish Ministry of Health.

Clinical assessments were conducted at Ankara University Medical School Hospital, one of the country's leading referral centers, renowned for its comprehensive medical expertise. As a government-subsidized institution within Türkiye's universal healthcare system, the hospital serves patients from diverse economic backgrounds. An analysis of the hometowns of the enrolled probands confirmed the diversity of participants, with individuals representing all regions of Türkiye. Moreover, control participants for the study were recruited from within the same families as the probands, ensuring comparable healthcare-seeking behavior between the affected and unaffected groups.

Patients admitted to the Neurology Department of Ankara University Medical School Hospital with the complaint of tremor were invited to participate in the genetic study if they reported having multiple affected and unaffected individuals in their families. Each proband underwent a comprehensive 90-minute baseline medical interview. If patients exhibited neurological symptoms, such as cognitive impairment, bradykinesia, or tremor, they were referred for detailed and follow-up neurological evaluations. We preferentially recruited familial early-onset ET cases, particularly those classified as "pure ET". Clinical information parameters included: a. type of tremor, b. presence or absence of Parkinson's disease, c. presence or absence of dystonia, d. response to treatment, e. history of drug use (e.g., lithium), f. substance use (e.g., alcohol or drug addiction), g. exposure to chemicals or toxins. Exclusion criteria were strictly applied to enhance the homogeneity of the study group and minimize confounding variables. We excluded individuals diagnosed with dementia, Parkinson's disease, or any other neurodegenerative disease, as well as those with known vascular, demyelinating, or structural brain lesions (e.g., masses). Additionally, individuals taking medications known to induce action tremor or those with severe hyperthyroidism were also excluded. However, recognizing the association between ET and a heightened risk of Parkinson's disease, patients with ET who exhibited overlapping parkinsonism features (e.g., bradykinesia, rigidity, and non-motor symptoms) were not excluded from the study [22–24]. Demographic information for each proband was collected during the initial interview. Probands were asked to provide detailed information about all living or deceased first- and second-degree relatives and indicate whether any relatives had tremors.

The accuracy of the pedigree information was critical for genetic analysis, and multiple checks were incorporated into the data collection process to ensure reliability and completeness. Recruitment of family members was coordinated during follow-up appointments, and efforts were made to have as many family members present as possible during these visits. This strategy facilitated the construction of robust pedigree charts, as family members could cross-verify information, minimizing discrepancies. During these appointments, family members collaborated to name and list all living and deceased first- and second-degree relatives, and they jointly reported on the presence of tremors in the family. This cross-referencing process ensured high specificity and accuracy in the familial data collected.

## Baseline evaluation

Ankara University Medical School physicians initially evaluated ET patients, and their family members based on clinical and functional performance tests. All clinical examinations were performed according to the Helsinki Declaration [25]. Each proband and family member recruited for the study was assigned a unique DNA code and subjected to comprehensive general physical and detailed neurological examination. The diagnostic criteria for ET were aligned with the Washington Heights-Inwood Genetic Study of Essential Tremor (WHIGET) [26] and the Consensus Statement of the Movement Disorder Society on Tremor (MDS) [27]. The severity of resting and postural tremors for each participant was graded from 0 to +3: 0, no visible tremors; +1, low-amplitude tremor; +2, moderate-amplitude tremor; and +3, high-amplitude tremor. To assess kinetic tremor, participants were examined while performing four different tasks: finger-to-nose movement, pouring water, drinking water, and drawing. ET patients were also evaluated for bradykinesia, muscular rigidity, and postural instability to assess the presence of Parkinson's disease or dystonia according to the UK Parkinson Disease Society Brain Bank diagnostic criteria [3,28]. This careful screening ensured that patients with overlapping parkinsonian features were appropriately identified and accounted for in the study.

During the clinical assessments, a trained research assistant collected demographic data (e.g., age, gender, race, and education level), as well as detailed information about the age of symptom onset, family history of ET, family history of Parkinson's disease, medication usage, and pedigree information. Enrollment of probands and their family members began in 2010 and was completed in 2019. Over this period, 161 families were initially recruited for the study; however, 16 families were excluded due to the inability to perform follow-up analysis resulting from a loss of contact. This exclusion ensured that the final dataset consisted only of participants with confirmed ET or individuals closely related to ET patients, with a final cohort drawn from 145 families (S1 Table).

Furthermore, it is worth noting that our study focused on individuals over the age of 40, as ET most commonly manifests after this age [14]. All in all, we collected clinical and demographic information from a total of 1,493 individuals aged 40 and over. The age of onset for ET was categorized into four distinct groups: a. early-onset (<30 years), b. intermediate-onset (30–59 years), c. late-onset (≥60 years), and d. onset unknown (where information was not collected). This approach provided a comprehensive dataset that allowed for the examination of potential correlations between age of onset, severity of symptoms, and genetic factors. The inclusion of both early-onset and late-onset cases further strengthened the study's ability to explore the genetic heterogeneity of ET across different age groups.

## Follow-up evaluation

Follow-up evaluations were conducted to update and expand upon the baseline data, ensuring the accuracy and completeness of the study's findings. Before commencing the

study, we successfully conducted follow-up assessments for 128 out of 161 probands (79%) through phone interviews. During these follow-ups, we updated information on pedigrees, demographic details (such as age, age of onset, and disease status), and vital status (alive or deceased) either from the probands or from their elderly relatives. Additionally, we supplemented the follow-up dataset for 17 out of 161 families (11%) by accessing medical records from healthcare providers. Due to the inability to maintain contact with certain families, 16 families (10%) were excluded from the study. As a result, we included 1,493 individuals, all aged 40 or older, consistent with the observation that the mean age of onset typically occurs around 40 years of age [29]. Within this cohort, 22 individuals had passed away since the baseline assessment. For those participants who were no longer available for direct examination, their ET status was determined using medical records obtained from general practitioners, hospitalization records, and neurologists (if they had been consulted).

The vital status (alive or deceased) and cause of death for each individual were systematically collected through family interviews. For deceased individuals, the ET status was reported by family members or verified by medical records if the individual had been diagnosed with ET by a specialist during their lifetime. Family members were also asked to assess the tremor severity of deceased individuals, comparing it with other relatives within the family to estimate the progression of symptoms. For further context, clinical evaluations and pedigree details for randomly selected two families are provided in the supplementary materials (see S2 Table and S1-S3 Figs in S1 File). These examples illustrate the familial patterns of ET, the progression of tremor severity, and demographic information relevant to our analyses.

In cases where family members or medical records provided the cause of death, a wide range of conditions was reported. These included natural deaths (a common description for older individuals, especially in earlier generations where precise medical diagnoses were unavailable, particularly in rural areas), heart attacks, strokes, cancers, respiratory diseases, infectious diseases, accidents, murders, and suicides. Since many of the reported causes of death in the control group were from earlier generations and lacked clinical specificity, "natural death" was frequently cited. In regions with limited access to healthcare, detailed post-mortem diagnoses were often not available, which posed a challenge in identifying exact causes of death in older records. To minimize bias in our analysis, we excluded individuals who died from early or accidental causes. This included deaths due to psychiatric disorders ($n = 8$), adoption where family genetic data could not be confirmed ($n = 3$), deaths at a relatively young age due to accidents or suicides ($n = 8$), deaths from infectious diseases (including COVID-19) ($n = 6$), deaths from cancer ($n = 1$), and deaths from unknown causes ($n = 7$). Excluding these individuals allowed us to maintain a more comparable distribution of age at death between ET+ (those with ET) and ET− (family members without ET). By removing these cases, we ensured that the analysis reflected age-related mortality due to common causes of death and reduced the influence of unrelated, non-age-related fatalities. This systematic approach provided a robust framework for assessing the association between ET and longevity and a clearer understanding of the causes of death in both the ET+ and ET− groups.

## Data analysis

All statistical analyses were performed using R (version 4.4.1) [30] and Python (version 3.11.5) [31], with survival analyses conducted via the lifelines library (version 0.27.8) [32]. The normality of the data was assessed using the Kolmogorov-Smirnov, Anderson-Darling, and Shapiro-Wilk tests. Additionally, visual inspections, including Q-Q plots and histograms, were employed to confirm data distribution. Given that the data did not meet the assumptions of normality, non-parametric statistical methods were applied.

Baseline characteristics of individuals with and without ET were compared using the non-parametric Mann-Whitney U test, which is suitable for analyzing non-normally distributed data. The Kruskal-Wallis test was used for group comparsions involving disease severity and age of onset, as it accommodates unequal sample sizes and non-normal data distributions. Post-hoc pairwise comparisons were performed using the Dunn test with Holm correction to account for multiple comparisons.

Survival analyses focused on two key outcomes: the 'age in years' for living participants and the 'age at death' for deceased individuals. Kaplan-Meier survival curves were constructed to estimate and compare survival probabilities between ET+ and ET− groups. Survival comparisons were stratified by cohort and gender to assess whether ET status influenced survival within these subgroups. The log-rank test was used to statistically compare survival curves, providing indication of whether the differences between groups were significant. To further assess the relationship between ET status and survival, univariate Cox proportional hazards regression models were applied. Univariate models evaluated the correlation between ET status and survival time, while multivariate models controlled for potential confounders, including gender and disease status. We used a stepwise model-building approach, sequentially introducing predictor variables and assessing their impact on survival, to ensure the most relevant prognostic factors were included in the final model [33]. Statistical significance was defined by $p$-value less than 0.05.

Additionally, to examine familial aggregation, ten families with at least ten members each were selected for individual analysis. Given the small sample sizes within these families, selecting the appropriate statistical method was crucial. We first used the Shapiro-Wilk test to assess the normality of age distributions within each family, with visual checks performed using Q-Q plots and histograms. For families where the data significantly deviated from normality, we applied the Mann-Whitney U test to compare age distributions. For those families with normally distributed data, comparisons of age distributions were conducted using unpaired t-test with Welch's correction. To complement the frequentist analysis, a Bayesian t-test was also performed, which provided additional insights into the comparison of age distributions between ET+ and ET− individuals within these families. Finally, the data from all ten families were combined and reanalyzed to evaluate overall trends. The combined data did not follow a normal distribution, as confirmed by the statistical tests and visual inspections. Thus, we employed the Kruskal-Wallis test to perform an overall comparison across families, which allowed us to evaluate differences in age distributions across multiple groups without assuming normality. This comprehensive approach ensured a robust evaluation of the association between ET and longevity, accounting for individual and familial variability.

## Results

### Baseline characteristics of the study cohort

The study included 1,493 participants from 145 families. Of these, 742 individuals were diagnosed with ET, while 751 family members were non-ET individuals. The median age of participants was 67 years, with an interquartile range (IQR) of 54–77 years. In the ET+ group, 264 individuals (133 females and 131 males) were deceased, with a median age of 80 years (IQR 70–86). Among the 478 living ET+ patients (228 females and 250 males), the median age was 63 years (IQR 53–74). In contrast, the ET− group had 334 deceased individuals (153 females and 181 males), with a median age of 70 years (IQR 59–77). Among the 417 living ET− individuals (246 females and 171 males), the median age was 60 years (IQR 49–71) (**Table 1**).

To select appropriate statistical tests, we performed three normality tests: Shapiro-Wilk test (W = 0.98, p = $1.11 \times 10^{-14}$), Kolmogorov-Smirnov test (D = 1, p < $2.2 \times 10^{-16}$), and

**Table 1. Baseline characteristics of the cohort.**

| Characteristic | ET− | | | ET+ | | |
|---|---|---|---|---|---|---|
| | **Alive** N = 417 | **Deceased** N = 334 | **Total** N = 751 | **Alive** N = 478 | **Deceased** N = 264 | **Total** N = 742 |
| **Age**[1] | 60 (49-71) | 70 (59-77) | 64 (52-75) | 63 (53-74) | 80 (70-86) | 70 (56-80) |
| **Age Ranges**[2] | | | | | | |
| 40-50 | 123 (29) | 42 (13) | 165 (22) | 101 (21) | 15 (5.7) | 116 (16) |
| 51-60 | 90 (22) | 57 (17) | 147 (20) | 112 (23) | 14 (5.3) | 126 (17) |
| 61-70 | 98 (24) | 89 (27) | 187 (25) | 111 (23) | 41 (16) | 152 (20) |
| 71-80 | 80 (19) | 99 (30) | 179 (24) | 98 (21) | 74 (28) | 172 (23) |
| 81-90 | 24 (5.8) | 39 (12) | 63 (8.4) | 51 (11) | 93 (35) | 144 (19) |
| 91-100 | 2 (0.5) | 8 (2.4) | 10 (1.3) | 5 (1.0) | 24 (9.1) | 29 (3.9) |
| >100 | 0 (0) | 0 (0) | 0 (0) | 0 (0) | 3 (1.1) | 3 (0.4) |
| **Gender**[2] | | | | | | |
| Female | 246 (59) | 153 (46) | 399 (53) | 228 (48) | 133 (50) | 361 (49) |
| Male | 171 (41) | 181 (54) | 352 (47) | 250 (52) | 131 (50) | 381 (51) |
| **ET Severity**[2] | | | | | | |
| Mild | NA | NA | NA | 91 (19) | 53 (20) | 144 (20) |
| Moderate | NA | NA | NA | 71 (15) | 29 (11) | 100 (14) |
| Severe | NA | NA | NA | 314 (66) | 179 (69) | 493 (67) |
| **Onset**[2] | | | | | | |
| Early | NA | NA | NA | 92 (27) | 22 (14) | 114 (23) |
| Intermediate | NA | NA | NA | 176 (51) | 60 (39) | 236 (47) |
| Late | NA | NA | NA | 76 (22) | 73 (47) | 149 (30) |
| **Parkinsonism Features**[2] | | | | | | |
| Yes | 0 (0) | 0 (0) | 0 (0) | 29 (6.1) | 14 (5.3) | 43 (5.8) |
| No | 417 (100) | 334 (100) | 751 (100) | 449 (94) | 250 (95) | 699 (94) |
| **Clinical Assessment**[2] | | | | | | |
| Yes | 73 (18) | 4 (1.2) | 77 (10) | 243 (51) | 14 (5.3) | 257 (35) |
| No | 344 (82) | 330 (99) | 674 (90) | 235 (49) | 250 (95) | 485 (65) |

[1]Median (IQR);

[2]n (%); NA, not applicable.

Anderson-Darling test (A = 8.55, p < $2.2 \times 10^{-16}$). The small p values in all tests indicated that the data were not normally distributed, prompting the use of non-parametric statistical methods to assess relationships between ET+ and ET− individuals.

To show the directionality of the association between ET status and aging, we compared the age distributions of ET+ and ET− individuals. We found a significant difference between two groups (Mann-Whitney U test, $U = 2.27 \times 10^5$, $p = 4.29 \times 10^{-10}$) (Fig 1A). When repeating the analysis in the elderly cohort (age ≥ 60 years), the results remained significant ($U = 9.29 \times 10^4$, $p = 5.61 \times 10^{-10}$) (Fig 1B).

We further evaluated the impact of ET severity and age of onset on age distributions. The Kruskal-Wallis test showed a significant increase in median ages as ET severity increased ($\chi2 = 28.63$, $df = 2$, $p = 6.08 \times 10^{-7}$) and as age of onset decreased ($\chi2 = 145.65$, $df = 2$, $p = 2.36 \times 10^{-32}$). Specifically, the median ages in severe, moderate, and mild ET cases were 72, 69, and 60, respectively. The median ages in early, intermediate, and late-onset ET cases were 56.5, 64, and 79, respectively. Post-hoc Dunn tests revealed significant differences between severe-moderate ($p = 8.14 \times 10^{-3}$) and severe-mild ($p = 5.74 \times 10^{-7}$) cases, though

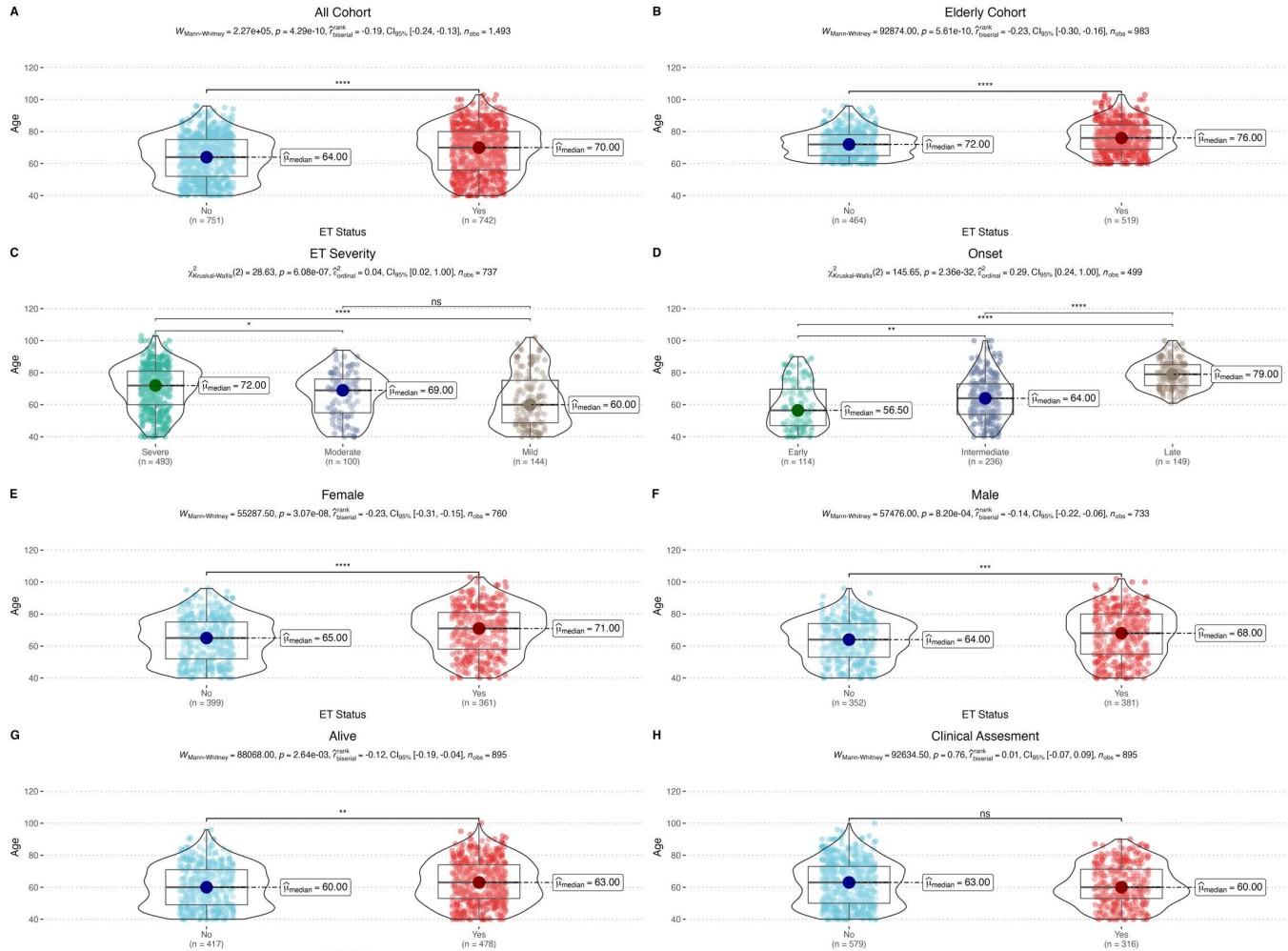

**Fig 1. Description of the cohort and comparison of median ages.** **(A)** Age distributions of ET+ and ET− individuals show a significant difference. **(B)** Analysis within the elderly cohort (age ≥ 60) confirms significant age differences between ET+ and ET− groups. **(C)** Increase in median ages with disease severity, with statistically significant pairwise comparisons. **(D)** Effect of age of onset on median ages, with all pairwise comparisons statistically significant. **(E)** Age distributions within females and **(F)** males exhibit significant differences based on ET status. **(G)** Median ages of living ET+ and ET− cases show a slight difference. **(H)** No significant difference in median age was observed between those with and without clinical assessment in the entire cohort. ( * $p < 0.05$, ** $p < 0.01$, *** $p < 0.001$, **** $p < 0.0001$, not significant (ns): $p > 0.05$).

no significant difference was observed between mild-moderate ($p = 0.07$) in ET severity. In contrast, all pairwise comparisons were significant for age of onset (early-intermediate, $p = 4.28 \times 10^{-3}$; early-late, $p = 1.38 \times 10^{-27}$; intermediate-late, $p = 5.18 \times 10^{-23}$) (Fig 1C-D). This significant increase in median ages with increasing ET severity was observed in both females ($U = 5.53 \times 10^{4}$, $p = 3.07 \times 10^{-8}$) (Fig 1E) and males ($U = 5.75 \times 10^{4}$, $p = 8.20 \times 10^{-4}$) (Fig 1F). A slight difference in the median ages was also noted between living ET+ and ET− cases ($U = 8.81 \times 10^{4}$, $p = 2.64 \times 10^{-3}$) (Fig 1G). In a comparison of the entire cohort, there was no significant difference in the median age between individuals with clinical assessment and those without ($U = 9.26 \times 10^{4}$, $p = 0.76$), indicating that clinical assessment did not bias the age distribution in our study (Fig 1H).

Afterward, we focused our analysis on deceased individuals. We compared the median age at death between ET+ and ET− individuals. We observed that ET+ cases lived significantly

longer, with a median age of 80 years, than ET– family members, who had a median age of 70 years ($U = 2.47 \times 10^4$, $p = 2.10 \times 10^{-20}$) (**Fig 2A**). Within the ET+ group, disease severity did not significantly affect survivals ($\chi 2 = 2.88$, $df = 2$, $p = 0.24$) (**Fig 2B**). However, the age of onset of ET significantly impacted survival ($\chi 2 = 14.87$, $df = 2$, $p = 5.90 \times 10^{-4}$). Post-hoc Dunn test pairwise comparisons showed significant differences between early-late ($p = 0.008$) and intermediate-late onset cases ($p = 0.002$), but not between early and intermediate onset ($p = 0.66$) (**Fig 2C**).

Lastly, we examined the relationship between gender and survival. A comparison of the mean age at death was observed between male and female ET+ and ET– individuals ($U = 5.29 \times 10^4$, $p = 8.07 \times 10^{-5}$) (**Fig 2D**). While clinical assessments did not influence survival patterns in the overall cohort, we observed a significant difference between clinically assessed individuals and those without clinical evaluations among the deceased participants ($U = 2.99 \times 10^3$, $p = 2 \times 10^{-3}$) (**Fig 2E**). This result aligns with expectations, as many first-generation individuals, particularly from earlier generations, were not clinically assessed during early times. On the other hand, the presence of parkinsonism (ET+PD+) did not significantly affect survival

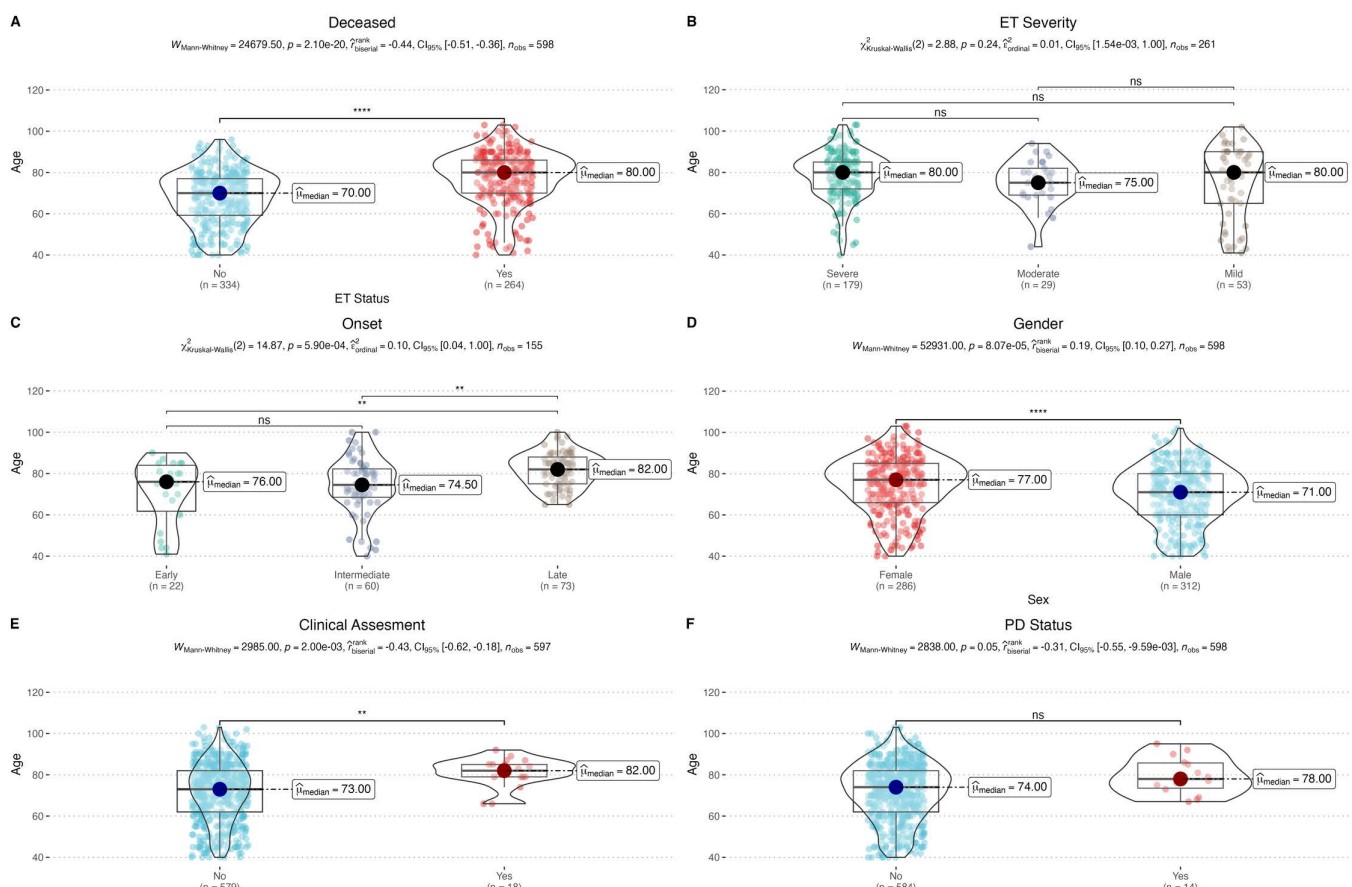

**Fig 2. Comparison of median ages at death.** **(A)** Median age at death is significantly higher in ET+ cases (80 years) compared to ET– family members (70 years). **(B)** No significant difference in disease severity among ET+ patients. **(C)** Median ages at death significantly differ by disease onset, with specific pairwise comparisons showing statistical significance. **(D)** Significant difference in median ages at death between ET+ and ET– individuals by gender. **(E)** Median ages at death differ significantly based on clinical assessment. **(F)** Parkinsonism status (ET+PD+) does not significantly affect survival patterns. (* $p < 0.05$, ** $p < 0.01$, *** $p < 0.001$, **** $p < 0.0001$, not significant (ns): $p > 0.05$).

patterns within the deceased cohort ($U$ = 2.838, $p$ = 0.05), suggesting that the overlap of ET and Parkinson's disease did not contribute to a noticeable difference in longevity. (**Fig 2F**).

## Survival analysis

We conducted a comprehensive survival analysis to compare the outcomes of ET+ and ET− individuals using Kaplan-Meier survival curves, which estimate overall survival probabilities over time. The log-rank test revealed significant differences in survival between ET+ and ET− individuals across the entire cohort, the elderly cohort, and when stratified by gender and ET status. The results were highly significant: $p = 1.11 \times 10^{-23}$ (Fig 3A); $p = 5.29 \times 10^{-16}$ (Fig 3B); $p = 3.88 \times 10^{-5}$ (Fig 3D); $p = 1.38 \times 10^{-16}$ (Fig 3E); $p = 1.00 \times 10^{-9}$ (Fig 3F). Notably, among only ET+ patients, the survival curves did not show a significant difference ($p$ = 0.054) (Fig 3C).

In the overall cohort, the median survival time for ET+ was 85 years, compared to 77 years for ET− individuals. In the elderly cohort, the median survival ages for ET+ and ET− individuals are 85 and 78 years, respectively. Stratifying by gender, ET+ males and females both had a median survival of 85 years. In contrast, ET− males had a median survival of 73 years, while ET− females had a median survival of 79 years. Similarly, among males, ET+ individuals had a median survival of 85 years compared to 73 years for ET−, while among females, ET+ individuals had a median survival of 85 years compared to 79 years for ET−.

To further explore the factors influencing survival, we performed univariate Cox regression analyses. ET status was found to significantly contribute to survival, with a hazard ratio (HR) of 0.44 ($CI_{95\%}$ = 0.37–0.52) in the overall cohort (Fig 3A and S3 Table in S1 File). Since aging is a risk factor for developing ET, we repeated the analysis in the elderly cohort (age ≥ 60 years), where a significant increase in survival time was also observed for ET+ patients compared to ET− individuals ($HR$ = 0.48, $CI_{95\%}$ = 0.40–0.58) (Fig 3B). In contrast, gender did not significantly influence survival in the ET+ cohort (HR = 1.28, $CI_{95\%}$ = 1.0–1.63) (Fig 3C), while it did in the ET− cohort ($HR$ = 1.57, $CI_{95\%}$ = 1.26–1.95) (Fig 3D). We then performed multivariate Cox regression analysis to adjust for potential confounding variables. These analyses confirmed that gender remained an independent predictor of survival within ET status. The adjusted hazard ratios for males were 0.66 ($CI_{95\%}$ = 0.47–0.93) for ET+ and 1.51 ($CI_{95\%}$ = 1.08–2.13) for ET− (**Fig 3E**). Among females, the adjusted hazard ratios were 0.73 ($CI_{95\%}$ = 0.52–1.02) for ET+ and 1.37 ($CI_{95\%}$ = 0.98–1.93) for ET− (**Fig 3F**).

## Comparison of mean age at death in families

To investigate whether ET+ cases in a single family lived longer than ET− relatives and in addition, to further scrutinize our clinical assessment methods for deceased individuals, we compared the mean ages at death of ET+ and ET− family members across 10 phenotypically well-characterized multigenerational families with multiple affected and unaffected individuals: Family-1, -17, -31, -32, -44, -45, -55, -104, -149 and -161.

First, we tested the normality of the age data for each family using the Shapiro-Wilk test. We found that age data from seven families followed a normal distribution (Family-17: $p$ = 0.66; Family-31: $p$ = 0.053; Family-32: $p$ = 0.45; Family-44: $p$ = 0.59; Family-55: $p$ = 0.25; Family-104: $p$ = 0.14; Family-149: $p$ = 0.85), while three families showed significant deviations from normality (Family-1: $p$ = 0.01; Family-45: $p$ = 0.018; Family-161: $p$ = 0.035). For families with normally distributed data, we compared mean ages at death using an unpaired t-test with Welch's correction and Bayesian t-test for the posterior distribution of ages. We used the non-parametric Mann-Whitney U test for families that did not meet the normality assumption. Overall, we found significant differences in age at death in three families: Family-44

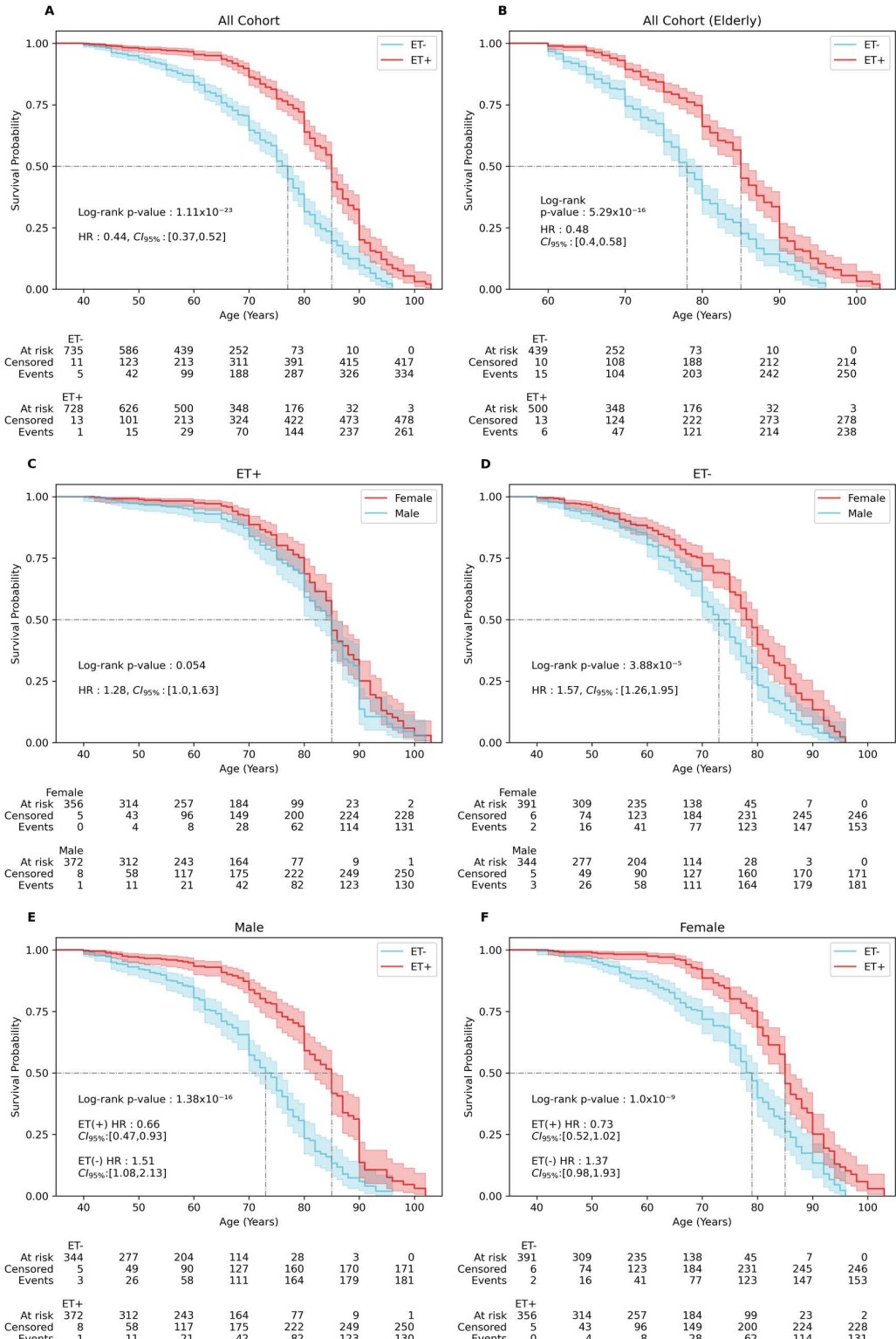

**Fig 3. Survival analysis of the cohort. (A)** Survival in patients with ET (red line) versus non-ET family members (blue line) in a univariate Cox Model, **(B)** Survival in ET (red line) versus non-ET (blue line) in a univariate Cox Model for elderly cohort, **(C)** Survival in females with (red line) versus males (blue line) in ET cohort in a univariate Cox Model, **(D)** Survival in females with (red line) versus males (blue line) in non-ET cohort in a univariate Cox Model, **(E)** Survival in males with ET (red line) and non-ET (blue line) in a multivariate Cox Model, **(F)** Survival in females with ET (red

line) and non-ET (blue line) in a multivariate Cox Model. All groups except ET male vs female family members were significantly different at $p < 0.0001$ with increased hazard ratios (HR).

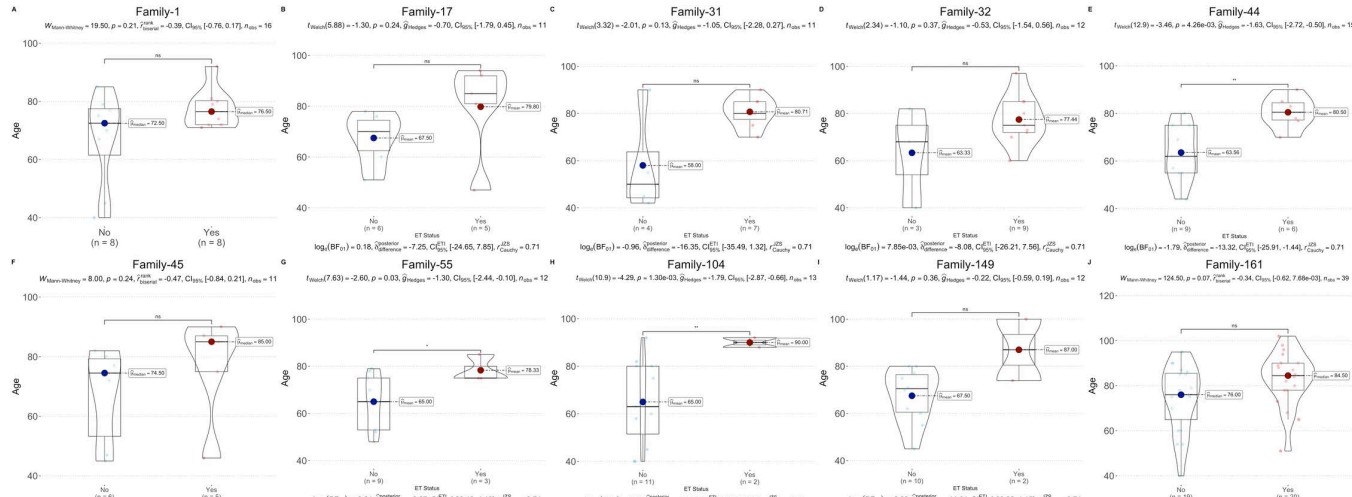

**Fig 4. Family-based analysis.** The median and mean ages of ET patients and non-ET family members in families **(A)** Family-1, **(B)** -17, **(C)** -31, **(D)** -32, **(E)** -44, **(F)** -45, **(G)** -55, **(H)** -104, **(I)** -149 and **(J)** -161 were tested individually. Mean and median ages in ET patients were higher in seven families, but the differences were not significant **(A)** Family-1, **(B)** -17, **(C)** -31, **(D)** -32, **(F)** -45, **(I)** -149, **(J)** -161. In three families **(E)** -44, **(G)** -55, **(H)** -104 differences were significant ( * $p < 0.05$, ** $p < 0.01$, *** $p < 0.001$, **** $p < 0.0001$, not significant (ns): $p > 0.05$).

($t_{welch}(12.9) = -3.46$, $p = 4.26 \times 10^{-3}$) (Fig 4E); Family-55 ($t_{welch}(7.63) = -2.60$, $p = 0.03$) (Fig 4G); and Family-104 ($t_{welch}(10.9) = -4.29$, $p = 1.30 \times 10^{-3}$) (Fig 4H). The remaining families did not show significant differences between ET+ and ET− groups (Family-1 ($U = 19.5$, $p = 0.21$) (Fig 4A); Family-17 ($t_{welch}(5.88) = -1.30$, $p = 0.24$) (Fig 4B); Family-31 ($t_{welch}(3.32) = -2.01$, $p = 0.13$) (Fig 4C); Family-32 ($t_{welch}(2.34) = -1.10$, $p = 0.37$) (Fig 4D); -45 ($U = 8.00$, $p = 0.24$) (Fig 4F); Family-149 ($t_{welch}(1.17) = -1.44$, $p = 0.36$) (Fig 4I); Family-161 ($U = 124.50$, $p = 0.07$) (Fig 4J)). Despite the lack of significant differences in most families, we observed a consistent trend where the mean and median ages at death were higher in the ET+ groups across all families. Given this observation, we combined the data from all 10 families and conducted a Kruskal-Wallis test due to the non-normal distribution of the overall data ($W = 0.95$, $p = 1.78 \times 10^{-5}$) This analysis confirmed that ET+ individuals had significantly longer lifespans than their ET− relatives ($\chi2 = 29.46$, $df = 2$, $p = 5.71 \times 10^{-8}$). In conclusion, 3 out of 10 families showed significant differences in the mean age at death, with ET+ individuals living longer than their ET− relatives, and a general trend toward longer lifespans for ET+ individuals was observed across all families.

## Discussion

Essential tremor (ET) is among the most prevalent movement disorders worldwide, yet its relationship with mortality and longevity remains an area of debate [34]. Our study adds to this complex body of literature by presenting evidence that ET may be associated with increased longevity. These findings, which align with earlier observations suggesting extended survival in ET patients, challenge conventional views of ET as a purely neurodegenerative condition.

The earliest evidence linking ET to longevity dates back to an observational study from the early 1900s, where Minor reported that one in every 14.5 (6.9%) patients with ET were 80 years old, compared to one in every 116 (0.9%) people in the general population of France [17]. He later noted, after observing the parents and grandparents of 51 ET cases, that "the older one is, the more likely one displays tremor," implying a possibility of a genetic factor involved in the tremor gamete for longevity [18]. This notion that ET might influence lifespan, or vice versa, went unverified for decades until Jankovic and colleagues in 1995 provided some of the first modern evidence for a potential link between ET and longevity. In their study, they observed that parents of ET patients with tremor lived on average, 9.2 years longer than those without tremor [20]. This observation hinted at potential anti-aging effects associated with ET. While intriguing, these early findings were based on relatively small samples and specific population groups, and the biological mechanisms underlying such an association were not well understood.

Subsequent studies on ET and mortality have been sparse, with only two major longitudinal studies addressing the topic. The first was a retrospective case-control study of 266 ET patients from Rochester, Minnesota, where researchers found that survival after the diagnosis of ET was comparable to age- and sex-matched controls from the general population of the West North Central United States. However, the study had several limitations: the participants were relatively young (mean age: 58 years), and they were not followed into advanced age (mean follow-up: 9.7 years). Thus, the study could not provide robust data on the long-term survival of ET patients, especially those who might develop age-related complications. The second major study was the prospective, population-based Neurological Disorders in Central Spain (NEDICES) cohort, which included 201 ET cases and 3,337 controls. This study found an increased risk of mortality among ET patients, with a risk ratio of 4.69 in an adjusted Cox model, and a p-value of less than 0.001, indicating a significant association between ET and higher mortality. However, the NEDICES study had its own limitations, including a relatively short follow-up period of just over three years, which may not have captured long-term survival trends. The conflicting results between these two studies highlight the complexity of studying mortality in ET and suggest that the relationship between ET and lifespan may vary based on study design, population characteristics, and other factors [19].

## Genetic contributions to longevity in ET

Genetic factors are likely to contribute significantly to the observed association between ET and increased lifespan. Several loci associated with ET, including those implicated in mitochondrial function and oxidative stress response, could influence both the development of tremor and the broader biological pathways governing aging. These genetic variations may provide protection against age-related diseases, such as cardiovascular conditions or metabolic syndromes, which are leading causes of mortality in the general population. Our family-based study design offers unique insights into the potential genetic basis of ET-related longevity. By comparing ET-positive (ET+) individuals to their ET-negative (ET−) relatives, we were able to control for many environmental and socioeconomic variables. By studying family members from the same households and communities, we sought to minimize these confounding variables. However, this design also introduces the potential for selection bias, as the families we studied may not represent the broader population. The genetic uniqueness of our cohort is underscored by an inbreeding coefficient of 36.5%, calculated from exome sequencing data, indicating a higher degree of genetic relatedness within these families than would be typical in the general population. The consistent trend of longer lifespans among ET+ individuals within the same families underscores the possibility that genetic factors influencing ET may

also contribute to longevity. Our findings provide important insights into the relationship between ET and mortality. We found that the median age of death for the ET+ group was significantly higher than for ET− individuals (80 years vs. 70 years), suggesting that ET might be associated with increased longevity. This was further supported by survival analysis, which revealed a statistically significant higher risk of mortality in ET− individuals compared to those with ET.

## Gender-specific survival patterns

Our study also identified gender-specific differences in survival among ET patients, with men showing a more pronounced longevity benefit than women. We observed notable gender-specific differences in survival. Males had a log-rank $p$-value of $1.38 \times 10^{-16}$ ($HR = 1.51$, $CI_{95\%}$ = 1.08–2.13), while females had a log-rank $p$-value of $1.0 \times 10^{-9}$ ($HR = 1.37$, $CI_{95\%}$ = 0.98–1.93), indicating that the longevity benefits of ET were more pronounced in men. This finding aligns with previous research highlighting gender-based differences in ET expression, such as the higher prevalence of head tremor in women and childhood-onset ET in men. Biological, hormonal, and behavioral factors likely interact to create these disparities. For instance, men and women differ in their susceptibility to certain comorbidities, such as cardiovascular disease, which could influence survival patterns in the context of ET. Additionally, gender differences in healthcare-seeking behavior and lifestyle choices may further contribute to these observations. Future studies should investigate these interactions to clarify the mechanisms behind gender-specific survival trends in ET [35,36].

When comparing our findings to national mortality data, we found that the median age of death in our cohort (74 years; males: 71, females: 77) was in line with the average life expectancy of the Turkish population in 2019, which was 78 years (males: 76, females: 81) [37]. However, when we focused on ET patients, their median age of death was 70 years (males: 68, females: 71), compared to 64 years (males: 64, females: 65) for their healthy relatives. While this suggests that ET patients in our study tended to live longer than their unaffected family members, we also found that the average year of death for deceased individuals in the ET− group was 2006 (1992–2021). This suggests that the time period during which deaths occurred may influence the observed differences in longevity, as life expectancy in Türkiye has increased over the past several decades.

One particularly intriguing finding from our study was that intermediate- and late-onset ET cases lived significantly longer than early-onset cases, by 7.5 and 22.5 years, respectively [10]. This finding aligns with prior research suggesting that late-onset ET, which is often observed in elderly populations, may have a different disease course compared to early-onset ET [10,21]. Aging-related tremor (ART) is increasingly recognized as a distinct entity from ET, with some studies suggesting that ART may be associated with normal aging processes rather than a neurodegenerative disorder [4]. The distinction between ET and ART is important for understanding the potential longevity benefits of ET, as ART may represent a milder form of tremor that is less likely to be associated with early mortality [38,39].

## Interpreting the longevity paradox in ET

The apparent longevity benefit in ET patients raises intriguing questions about the biological and clinical underpinnings of this association. While ET is traditionally considered a neurodegenerative disorder, its classification is evolving. Growing evidence suggests that ET represents a spectrum of conditions rather than a singular disease entity [40]. Subtypes of ET may differ in their underlying mechanisms, ranging from degenerative processes involving the cerebellum to milder, age-related phenomena. This heterogeneity could mean that specific

forms of ET are associated with less severe neurodegeneration or even adaptive changes that confer survival advantages [8]. The heterogeneity of ET, highlighted in neuroimaging and pathological studies, supports the idea that not all ET cases involve the same degree of neuro-degeneration [8].

Addionally, it is possible that in some ET patients, compensatory neural mechanisms mitigate the effects of neurodegeneration. This could preserve neurological function and contribute to a longer lifespan. However, further studies are needed to explore this hypothesis in detail, particularly using advanced neuroimaging techniques to assess adaptive neural changes in ET [41].

Several studies have identified genetic variations associated with ET that could be linked to pathways involved in aging and longevity. For instance, mutations in genes related to mitochondrial function or oxidative stress response pathways could influence both the development of ET and the lifespan of affected individuals [39,42,43]. Additionally, familial aggregation of ET may suggest the presence of genetic modifiers that influence not only tremor development but also overall health and longevity [7]. Some of these genetic factors may confer resilience to other age-related conditions, such as cardiovascular or metabolic diseases, which are common causes of mortality in the general population. For instance, genetic studies have identified specific loci associated with ET, including LINGO1 and STK32B, which might also influence longevity [7]. These protective genetic traits could help explain the observed association between ET and increased lifespan.

One more plausible explanation for the increased longevity observed in our study could be improved medical compliance among individuals with ET. Chronic conditions such as ET often require regular medical check-ups and adherence to prescribed therapies, which could translate into better management of comorbidities and early detection of other health issues. This is consistent with findings from other studies, which have shown that individuals with chronic diseases tend to seek more frequent medical care and engage in healthier lifestyle practices that contribute to prolonged life. Furthermore, patients with ET may be more likely to adopt preventive measures such as regular exercise or medications, which could contribute to their extended lifespan [44–46].

Environmental and lifestyle factors, including diet, physical activity, and alcohol consumption, could also play a role in the longevity of ET patients. Previous research has indicated that moderate alcohol consumption may have neuroprotective effects, potentially alleviating tremor symptoms and improving motor function [46,47]. Exercise, which has been shown to improve cognitive function and overall health, could also have a beneficial effect on individuals with ET, mitigating some of the negative consequences of aging [48,49]. Furthermore, dietary factors such as caloric restriction have been linked to extended lifespan in both animal models and human populations, and it is possible that individuals with ET may adopt diets that reduce inflammation and oxidative stress, both of which contribute to aging [7,43].

Lastly, it is important to consider the potential for selection bias in studies reporting enhanced longevity in ET patients. For instance, individuals with milder ET forms who live longer may be more likely to participate in research, which could skew the results [1]. Careful design of future studies is essential to address this limitation. Given the surprising nature of these findings, further research is critical. Longitudinal cohort studies with detailed genetic profiling and neuroimaging data could help elucidate the biological mechanisms underlying the association between ET and increased lifespan [50].

## Limitations and challenges

Despite its strengths, our study has limitations that warrant consideration. First, 35.6% of ET+ cases and 44.3% of ET− individuals were deceased family members, and 22 individuals had died after the baseline evaluation. For many of these individuals, assessments were based on

family-reported histories rather than direct clinical evaluations, which introduces the potential for recall bias and misclassification of ET status. To mitigate this, we compared survival data between ET patients with and without clinical assessments, finding a significant difference in age at death between the two groups ($U = 2.99 \times 10^3$, $p = 2 \times 10^{-3}$) (Fig 2E). Furthermore, we examined 10 phenotypically well-characterized multigenerational families, finding that in 3 of the 10 families, ET patients lived longer than their non-ET relatives. This finding suggests that genetic factors contributing to ET may also play a role in influencing longevity [51]. Additionally, the lack of detailed cause-of-death information, particularly for older individuals or those from rural areas, limits our ability to explore specific mortality patterns in depth. Deaths categorized as "natural causes" often lack specificity, potentially obscuring critical insights into comorbidities and other factors influencing survival. Access to more precise cause-of-death data, such as death certificates or autopsy reports, would provide valuable insights into whether specific comorbidities or conditions, such as cardiovascular disease or neurodegenerative disorders, are common in ET patients compared to the general population. Furthermore, our exclusion of individuals who died from non-age-related causes (e.g., accidents or suicides) may introduce bias by narrowing the study population to those with age-related mortality. While this approach was necessary to focus on longevity, it may limit the generalizability of our findings. We cannot entirely rule out this possibility, however our analysis of individuals aged 60 and older still demonstrated a significant increase in survival times for ET patients (Log-rank $p = 5.29 \times 10^{-16}$ (Fig 3B)), suggesting that the longevity benefit associated with ET is robust even among older individuals. Additionally, our study did not account for certain lifestyle factors, such as diet, physical activity, and alcohol consumption, which may influence both ET development and longevity. Further studies that incorporate detailed data on these factors would help to clarify their potential impact.

## Future directions

To build on these findings, future research should incorporate larger, more diverse cohorts and longitudinal follow-ups with detailed cause-of-death data. Advanced genetic analyses, including whole-genome sequencing, could provide deeper insights into the molecular pathways linking ET to longevity. Additionally, neuroimaging studies may help elucidate whether compensatory neural mechanisms contribute to the observed survival advantage. One potential biological mechanism is the neuroprotective effect of ET. Although ET is traditionally considered a movement disorder, there is emerging evidence suggesting that certain pathophysiological aspects of the disease may confer protective effects on the nervous system. Recent studies have proposed that the tremor may be related to enhanced neural plasticity, which could play a role in improving neuroprotection and mitigating neurodegeneration in certain brain regions. For example, it has been suggested that the constant neuronal activity associated with tremor may induce adaptive changes that enhance resilience to age-related neurodegeneration [52]. This could potentially explain why individuals with ET might experience slower cognitive decline or improved survival compared to the general population. Another potential mechanism involves the dopaminergic system. While ET is not typically classified as a neurodegenerative disorder like Parkinson's disease, it has been suggested that some individuals with ET may have compensatory mechanisms involving dopamine or other neurochemicals that confer a neuroprotective effect [53]. Studies have indicated that the basal ganglia circuitry, which is implicated in motor control and is disrupted in tremor, may also have a role in regulating other neurochemical pathways that influence aging and longevity. For example, dopamine's role in modulating oxidative stress and inflammation could play a key role in enhancing cellular repair mechanisms and slowing the aging process [42,43,54].

## Conclusion

In summary, our study adds to the limited body of literature on ET and mortality by suggesting that ET may be associated with increased lifespan, particularly in male patients and those with late-onset disease. However, our findings should be interpreted with caution due to the study's limitations, including the reliance on self-reported family histories and the lack of detailed cause-of-death data. Future research should aim to collect more comprehensive biological, genetic, and lifestyle data, as well as to include larger and more diverse cohorts, to validate and extend these findings. Clarifying the mechanisms underlying the relationship between ET and longevity could yield important insights into both ET pathophysiology and the broader biological processes that contribute to healthy aging.

## Contribution to the field statement

Essential tremor is one of the most frequent human movement disorders. There are surprisingly few data on mortality risk for ET in the literature. The debate on the effect of ET on longevity started in the early 1900s. Firstly, Minor claimed that 6.9% of his patients with ET were more than 80 years of age, significantly increasing compared to the general population (0.9%) in 1922. Seven decades later, in 1995, Jankovic reported that patients with ET had lived a median of 8 years longer than did parents who had not had tremor. There has been only one longitudinal retrospective case-control study of 266 patients with ET. The authors concluded that survival after ET was comparable to the expected survival for persons of similar age and sex. However, in that study, the participants were quite young, suggesting that some of the cases may not have been followed into advanced age when the risk of mortality is likely to be higher. We conducted a retrospective study that compared patients with ET to their relatives without ET in families. We reported that the individuals with ET had a significantly higher mean age of death and survival probability than their healthy relatives and the general population. We believe this study will motivate neurologists for prospective studies for groups of patients with essential tremor with respect to longevity and molecular biologists to understand the underlying causal link between longevity and essential tremor.

## Supporting information

**S1 Table. Demographics of the cohort.** The table includes demographic information, including age, gender, disease status (ET&PD), clinical assessment status, disease onset and severity, living status, relationship with the proband, and listing criteria from 1,493 individuals. (XLSX)

**S1 File. Supplementary materials.** This file contains S1-S3 Figs and S2 and S3 Tables. (PDF)

## Acknowledgements

We sincerely thank Prof. Tayfun Özçelik for his invaluable insights and constructive feedback on the manuscript. We would like to thank the volunteers for participating in the study.

## Author contributions

**Conceptualization:** Onur Emre Onat, Kivilcim E. Doganyigit, Merve Sen, Kaya Bilguvar, Muhittin Cenk Akbostanci.

**Data curation:** Onur Emre Onat, Cem Akbostanci, Kivilcim E. Doganyigit, Merve Sen, Emre Can Gunaydin, Muhittin Cenk Akbostanci.

**Formal analysis:** Onur Emre Onat, Faruk Ustunel, Cem Akbostanci, Kivilcim E. Doganyigit, Merve Sen.

**Investigation:** Onur Emre Onat, Muhittin Cenk Akbostanci.

**Methodology:** Onur Emre Onat, Cem Akbostanci, Kivilcim E. Doganyigit, Merve Sen, Muhittin Cenk Akbostanci.

**Project administration:** Onur Emre Onat.

**Software:** Faruk Ustunel.

**Supervision:** Onur Emre Onat.

**Validation:** Onur Emre Onat, Faruk Ustunel.

**Visualization:** Faruk Ustunel.

**Writing – original draft:** Onur Emre Onat, Faruk Ustunel, Kaya Bilguvar.

**Writing – review & editing:** Onur Emre Onat, Faruk Ustunel, Merve Sen, Kaya Bilguvar, Muhittin Cenk Akbostanci.

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
