## [Decision Letter · Decision Letter 0]

15 Aug 2024

PONE-D-24-19093Effects of essential tremor on longevity and mortality rates in familiesPLOS ONE

Dear Dr. Onat,

Thank you for submitting your manuscript to PLOS ONE. After careful consideration, we feel that it has merit but does not fully meet PLOS ONE’s publication criteria as it currently stands. Therefore, we invite you to submit a revised version of the manuscript that addresses the points raised during the review process.

We look forward to receiving your revised manuscript.

Kind regards,

Muhammad Zulkifl Hasan, PhD

Academic Editor

PLOS ONE

Journal Requirements:

"The authors declare that the research was conducted without any commercial or financial relationships that could be construed as a potential conflict of interest."

5. We are unable to open your Supporting Information file [S2 and S3 File]. Please kindly revise as necessary and re-upload.

Additional Editor Comments:

The manuscript provides valuable insights into the relationship between essential tremor (ET) and longevity. However, there are areas where improvements can enhance the impact of the work. First, the introduction could better contextualize the study by clearly defining the research gaps and how this study specifically addresses them. Additionally, the methodology section, while comprehensive, would benefit from further elaboration on data collection procedures, particularly the criteria used to select family members for inclusion. Enhancing the clarity of the statistical methods used would also help readers follow the analysis more easily. Moreover, the discussion could delve deeper into the potential biological mechanisms linking ET to longevity, as this connection remains speculative. Strengthening this section with references to recent studies in the field could offer a more robust interpretation of the findings. Lastly, a thorough proofreading of the manuscript is necessary to correct minor grammatical errors and ensure clarity.

Reviewers' comments:

Reviewer's Responses to Questions

**Comments to the Author**

1. Is the manuscript technically sound, and do the data support the conclusions?

Reviewer #1: Yes

2. Has the statistical analysis been performed appropriately and rigorously? 

Reviewer #1: Yes

3. Have the authors made all data underlying the findings in their manuscript fully available?

Reviewer #1: Yes

4. Is the manuscript presented in an intelligible fashion and written in standard English?

Reviewer #1: Yes

5. Review Comments to the Author

Reviewer #1: Small Sample Size: The study’s sample size, though substantial, is still limited. The authors acknowledge that more extensive studies are necessary to confirm the findings and understand the broader implications.

Family-based Study Limitations: By focusing on families, the study may have introduced selection bias. The family-based design might not represent the general population accurately, and the genetic and environmental factors specific to these families may not be generalizable.

Data Collection Methodology: The study relies on historical data provided by participants and their families, which could introduce recall bias and inaccuracies. Many assessments were based on reports rather than direct examinations, which can lead to misclassification of ET status.

Lack of Detailed Cause of Death Data: For many deceased individuals, the exact cause of death was not known and was broadly classified as "natural death," which could obscure specific causes that might influence longevity.

Survival Analysis: The paper mentions potential biases in survival analysis, such as the possibility that individuals who lived longer were more likely to develop ET, rather than ET contributing to longer life. This reverse causation could affect the interpretation of results.

Confounding Variables: Although the study attempts to control for certain variables, there are many potential confounders (e.g., lifestyle factors, comorbidities) that were not accounted for, which could influence the association between ET and longevity.

Gender Differences: The study finds gender-specific differences in survival rates, suggesting that males with ET might have a survival advantage. However, the underlying reasons for this gender disparity are not explored in depth, leaving questions about the mechanisms driving these differences.

Speculative Interpretations: The authors present several speculative explanations for their findings (e.g., dietary habits, medical compliance, alcohol consumption, exercise benefits), but these are not backed by solid biological data within the study, requiring further investigation.

Potential Misclassification: The study includes ET patients with overlapping parkinsonism features, which could complicate the classification and analysis. The exclusion criteria were stringent, but the presence of non-motor symptoms and other conditions might still affect the results.

6. PLOS authors have the option to publish the peer review history of their article (what does this mean? ). If published, this will include your full peer review and any attached files.

**Do you want your identity to be public for this peer review?** For information about this choice, including consent withdrawal, please see our Privacy Policy .

Reviewer #1: **Yes: ** Muhammad Zunnurain Hussain

---

## [Author Response · Author response to Decision Letter 1]

19 Oct 2024

Manuscript PONE-D-24-19093: Response to Reviewers

Dear Dr. Hasan,

Thank you for providing us the opportunity to submit a revised draft of our manuscript titled “Effects of Essential Tremor on Longevity and Mortality Rates in Families” for publication in PLOS ONE. We sincerely appreciate the time and effort you and the reviewers have dedicated to evaluating our work. The insightful comments and valuable suggestions we received have greatly contributed to improving the quality of our manuscript.

We have carefully considered all the feedback provided, and we have incorporated most of the reviewers’ suggestions into the revised manuscript. In response to your guidance, we rewrote the entire text and utilized the track changes feature in Word to clearly highlight all modifications. Additionally, we have prepared a clean version of the manuscript for your convenience. Below, we provide a point-by-point response to the reviewers’ comments and concerns in blue font. Please note that all page numbers mentioned correspond to the revised manuscript without tracked changes.

With my best regards,

Onur Emre Onat

Journal Requirements:

Author Response: We have carefully reviewed the style templates and revised the entire manuscript to fully adhere to the “PLOS ONE” formatting guidelines.

2. Thank you for stating the following in the Competing Interests section: "The authors declare that the research was conducted without any commercial or financial relationships that could be construed as a potential conflict of interest.” Please confirm that this does not alter your adherence to all PLOS ONE policies on sharing data and materials, by including the following statement: "This does not alter our adherence to PLOS ONE policies on sharing data and materials.” (as detailed online in our guide for authors http://journals.plos.org/plosone/s/competing-interests). If there are restrictions on sharing of data and/or materials, please state these. Please note that we cannot proceed with consideration of your article until this information has been declared. Please include your updated Competing Interests statement in your cover letter; we will change the online submission form on your behalf.

Author Response: We have included the following statement, which was previously categorized under the “Conflict of Interest Statement” section, now updated to “Competing Interests.

Page 17, Lines 611-614

The authors declare that the research was conducted without any commercial or financial relationships that could be construed as a potential conflict of interest. This does not alter our adherence to PLOS ONE policies on sharing data and materials.

Author Response: Data previously referred to as “data not shown” were not core to the research presented in our study; therefore, we have removed this phrase.

Author Response: We have relocated and incorporated our ethics statement into the Methods section, as outlined in the following paragraph.

Page 4, Lines 105-113

This study is a part of an ongoing project “Genetics of Essential Tremor in Turkish Families: Identification of the Causal Variants.” The primary aim of the study is to explore the genetic underpinnings of essential tremor (ET) in familial cases across Turkey. Written (signed) informed consent was obtained from all participants at the time of enrollment. All procedures involving human participants adhered to the ethical standards set by the institutional and national research committees, as well as the 1964 Helsinki Declaration and its later amendments, or comparable ethical guidelines. Ethical approval for the study was obtained from the Institutional Review Boards of both Bilkent University and Ankara University, under the regulations and guidelines issued by the Ministry of Health, Turkey.

5. We are unable to open your Supporting Information file [S2 and S3 File]. Please kindly revise as necessary and re-upload.

Author Response: The Supporting Information files, S2 and S3, contain the Python and R codes used in this study. These codes have been uploaded to a GitHub repository, with the links provided in the “Data Availability Statement” section, in accordance with PLOS ONE’s “Code Sharing” policies. We updated this section as in the following paragraph:

Page 17, Lines 631-636

Data are available in the main text or the supplementary materials. The raw data supporting the conclusions of this article will be made available by the authors without undue reservation. The R Markdown and Python codes used for the analysis are accessible in the GitHub repository (https://github.com/farukustunel/et-longevity-analysis). These codes include scripts for data processing, statistical analysis, and visualization.

Additional Editor Comments:

The manuscript provides valuable insights into the relationship between essential tremor (ET) and longevity. However, there are areas where improvements can enhance the impact of the work.

Author Response: We have revised the entire manuscript and rewritten all sections based on your valuable feedback.

First, the introduction could better contextualize the study by clearly defining the research gaps and how this study specifically addresses them.

Author Response: We revised the introduction part to clearly contextualize the text and addressed the research gaps as in the following paragraph:

Page 3, Lines 87-102

Building on this existing body of research, with contradictory results, further investigation into the relationship between ET and mortality is crucial. In our family-based retrospective study, we aimed to address this gap by comparing mortality risk in ET patients to their family members. We examined the pedigrees of 145 probands and tested the association of ET with longevity in 1,493 subjects. Our findings suggest that ET could indeed be associated with longer life expectancy, offering an intriguing avenue for further research into the underlying mechanisms that might confer this potential longevity advantage. These findings, if confirmed in large-scale longitudinal studies and genetic investigations, could have significant implications for public health, particularly in understanding the complex interplay between ET, aging and survival.

Additionally, the methodology section, while comprehensive, would benefit from further elaboration on data collection procedures, particularly the criteria used to select family members for inclusion.

Author Response: We revised the whole methods section, including the data collection procedures as in the following paragraph:

Page 3-4, Lines 114-151

The clinical assessments were conducted at the Ankara University Medical School Hospital, one of the country’s leading referral centers, renowned for its comprehensive medical expertise. As a government-subsidized institution within Turkey’s universal healthcare system, the hospital serves patients from diverse economic backgrounds. An analysis of the hometowns of the enrolled probands confirmed the diversity of participants, with individuals representing all regions of Türkiye. Moreover, control participants for the study were recruited from within the same families as the probands, ensuring comparable healthcare-seeking behavior between the affected and unaffected groups.

Patients admitted to the Neurology Department of Ankara University Medical School Hospital with the complaint of tremor were invited to participate in the genetic study if they reported having multiple affected and unaffected individuals in their families. Each proband underwent a comprehensive 90-minute baseline medical interview. If patients exhibited neurological symptoms, such as cognitive impairment, bradykinesia, or tremor, they were referred for detailed and follow-up neurological evaluations. We preferentially recruited familial early-onset ET cases, particularly those classified as “pure ET”. Clinical information parameters included: a. type of tremor, b. presence or absence of Parkinson’s disease, c. presence or absence of dystonia, d. response to treatment, e. history of drug use (e.g., lithium), f. substance use (e.g., alcohol or drug addiction), g. exposure to chemicals or toxins. Exclusion criteria were strictly applied to enhance the homogeneity of the study group and minimize confounding variables. We excluded individuals diagnosed with dementia, Parkinson's disease, or any other neurodegenerative disease, as well as those with known vascular, demyelinating, or structural brain lesions (e.g., masses). Additionally, individuals taking medications known to induce action tremor or those with severe hyperthyroidism were also excluded. However, recognizing the association between ET and a heightened risk of Parkinson’s disease, patients with ET who exhibited overlapping parkinsonism features (e.g., bradykinesia, rigidity, and non-motor symptoms) were not excluded from the study [22–24]. Demographic information for each proband was collected during the initial interview. Probands were asked to provide detailed information about all living or deceased first- and second-degree relatives and indicate whether any relatives had tremors.

The accuracy of the pedigree information was critical for genetic analysis, and multiple checks were incorporated into the data collection process to ensure reliability. Recruitment of family members occurred during follow-up appointments, and efforts were made to have as many family members present as possible during these visits. This strategy facilitated the construction of robust pedigree charts, as family members could cross-verify information, minimizing discrepancies. During these appointments, family members collaborated to name and list all living and deceased first- and second-degree relatives, and they jointly reported on the presence of tremors in the family. This cross-referencing process ensured high specificity and accuracy in the familial data collected.

Enhancing the clarity of the statistical methods used would also help readers follow the analysis more easily.

Author Response: To enhance the clarity of the statistical methods we used, we revised the statistical methods section in a more structured and detailed manner, ensuring that each step of our analysis is clearly explained.

Page 7-8, Lines 228-268

Statistical analyses were conducted using R version 4.4.1 [30], while survival analyses were performed in Python version 3.11.5 [31], utilizing the lifelines library version 0.27.8 [32]. To assess the normality of the data, several tests were applied, including the Kolmogorov-Smirnov test, Anderson-Darling test, and Shapiro-Wilk test. In addition, the data distribution was visually examined using normal quantile-quantile (Q-Q) plots and a histogram.

The baseline characteristics of individuals with and without ET were compared using the non-parametric Mann-Whitney U test, which is appropriate for non-normally distributed data. Additionally, the Kruskal-Wallis test was used to compare groups regarding disease severity and age of onset, as this test is suited for unequal sample sizes and non-normal data distributions. To determine specific differences among groups, a post-hoc Dunn test was applied with Holm correction to adjust for multiple comparisons.

For survival analysis, the main outcome of interest was ‘age in years’ for living individuals and ‘age-at-death’ for deceased individuals. Kaplan-Meier survival curves were constructed to estimate and compare survival probabilities between ET+ and ET- groups. Survival comparisons were stratified by cohort and gender to assess whether ET status influenced survival within these subgroups. The log-rank test was performed to statistically compare survival curves, providing indication of whether the differences between groups were significant. To explore the relationship between ET status and overall survival, univariate Cox proportional hazards regression models were employed. This allowed us to assess the correlation between disease status (ET+ or ET-) and survival time. A multivariate Cox regression model was then built to account for potential confounders such as gender, while still assessing the effect of disease status. We used a stepwise model-building approach, sequentially introducing predictor variables and assessing their impact on survival, to ensure the most relevant prognostic factors were included in the final model [33]. Statistical significance was defined by p-value less than 0.05

To provide additional insight into the familial aggregation, we selected 10 families, each consisting of at least 10 individuals, for family-wise comparisons. Given the small sample sizes within these families, selecting the appropriate statistical method was crucial. We first used the Shapiro-Wilk test to assess the normality of age distributions within each family. Visual checks using Q-Q plots and histograms were also conducted for further confirmation. For families where the data significantly deviated from normality, we applied the Mann-Whitney U test to compare age distributions. For those families where the data appeared normally distributed, an unpaired t-test with Welch’s correction was used to account for unequal variances. To complement the frequentist analysis, a Bayesian t-test was also performed, which provided additional insights into the comparison of age distributions between ET+ and ET- individuals within these families. After analyzing the families individually, we combined the data from all 10 families and re-evaluated the overall distribution of ages. The combined data did not follow a normal distribution, as confirmed by the statistical tests and visual inspections. Thus, we employed the Kruskal-Wallis test to perform an overall comparison across families, which allowed us to evaluate differences in age distributions across multiple groups without assuming normality.

Moreover, the discussion could delve deeper into the potential biological mechanisms linking ET to longevity, as this connection remains speculative. Strengthening this section with references to recent studies in the field could offer a more robust interpretation of the findings.

Author Response: In the revised manuscript, we have largely removed speculative sentences and presented potential biological mechanisms in a more cautious and nuanced manner. We have also clarified that these explanations are preliminary and should be considered as hypotheses rather than definitive conclusions. Additionally, we have expanded the discussion to emphasize the need for further research to investigate these potential factors through more rigorous data collection and analysis. Please see following:

Page 15-16, Lines

---

## [Decision Letter · Decision Letter 1]

29 Dec 2024

PONE-D-24-19093R1Effects of essential tremor on longevity and mortality rates in familiesPLOS ONE

Dear Dr. Onat,

Thank you for submitting your manuscript to PLOS ONE. After careful consideration, we feel that it has merit but does not fully meet PLOS ONE’s publication criteria as it currently stands. Therefore, we invite you to submit a revised version of the manuscript that addresses the points raised during the review process.

We look forward to receiving your revised manuscript.

Kind regards,

Amina Nasri

Academic Editor

PLOS ONE

Reviewers' comments:

Reviewer's Responses to Questions

**Comments to the Author**

1. If the authors have adequately addressed your comments raised in a previous round of review and you feel that this manuscript is now acceptable for publication, you may indicate that here to bypass the “Comments to the Author” section, enter your conflict of interest statement in the “Confidential to Editor” section, and submit your "Accept" recommendation.

Reviewer #2: All comments have been addressed

Reviewer #3: (No Response)

2. Is the manuscript technically sound, and do the data support the conclusions?

Reviewer #2: Yes

Reviewer #3: No

3. Has the statistical analysis been performed appropriately and rigorously? 

Reviewer #2: Yes

Reviewer #3: No

4. Have the authors made all data underlying the findings in their manuscript fully available?

Reviewer #2: Yes

Reviewer #3: (No Response)

5. Is the manuscript presented in an intelligible fashion and written in standard English?

Reviewer #2: Yes

Reviewer #3: Yes

6. Review Comments to the Author

Reviewer #2: The aim of this study is to analyze the effects of essential tremor on longevity and mortality rates in families. It is very interesting in all parts. I suggest to revise the language.

Reviewer #3: In this study Onur Emre Onat et al. investigated the potential link between Essential Tremor (ET) and longevity. They included 1,493 participants from 145 families, divided into ET-positive (ET+) and ET-negative (ET-) groups. The analysis aimed to clarify whether ET status is associated with lifespan. Results showed that the median age at death for deceased ET+ individuals was 80 years, compared to 70 years for ET- individuals. Living ET+ individuals had a median age of 63 years, while ET- individuals had a median age of 60 years. Survival analysis revealed a significantly longer lifespan for the ET+ group, with a hazard ratio of 0.44, indicating a lower mortality risk in ET+ patients, particularly among males.

I believe the study presents several methodological weaknesses that undermine the reliability and validity of its conclusions:

Specific comments:

- The study relies heavily on retrospective data for deceased patients, reconstructed through family trees. This approach introduces substantial recall bias, particularly regarding the accuracy of death diagnoses, age at death, and the presence of tremor. The reliance on family-reported data significantly undermines the validity and reliability of these key

- The inclusion of participants from the same families raises concerns about genetic and environmental confounding. Family-based sampling does not control for hereditary factors that could influence both tremor and longevity. The diversity in socio-economic backgrounds within the sample is insufficient to ensure the external validity of the results, as it does not reflect the heterogeneity of the general population.

- The study does not sufficiently address the potential inclusion of patients exhibiting parkinsonian features within the ET cohort. Without clear diagnostic criteria distinguishing ET or ET plus from Parkinson's disease (PD), there is a risk of misclassification, which could substantially alter the study's outcomes. Moreover, the absence of a comprehensive follow-up to confirm the absence of PD or other neurodegenerative conditions raises further concerns about diagnostic accuracy.

- The results, which suggest enhanced longevity in ET patients, are contrary to the possible understanding of ET as a neurodegenerative condition. The study does not adequately explore the potential biological mechanisms that could explain this surprising association between ET and increased lifespan. Given the neurodegenerative nature of the disorder, the authors should provide a more detailed discussion of the underlying pathophysiology that could account for the observed mortality differences, as this remains an unexplained and significant gap with respect to the current literature.

7. PLOS authors have the option to publish the peer review history of their article (what does this mean? ). If published, this will include your full peer review and any attached files.

**Do you want your identity to be public for this peer review?** For information about this choice, including consent withdrawal, please see our Privacy Policy .

Reviewer #2: No

Reviewer #3: No

---

## [Author Response · Author response to Decision Letter 2]

20 Jan 2025

Manuscript PONE-D-24-19093R1

Responses to the Reviewers' comments:

Reviewer #2: The aim of this study is to analyze the effects of essential tremor on longevity and mortality rates in families. It is very interesting in all parts. I suggest to revise the language.

Author Response: We sincerely thank the reviewer for their positive feedback and for highlighting the importance of our study. In response to the suggestion to revise the language, we have carefully reviewed and refined the manuscript to enhance its clarity and readability. Additionally, we have incorporated new content and reorganized key sections to improve the overall narrative and ensure the study’s objectives and findings are presented more effectively.

Reviewer #3: In this study Onur Emre Onat et al. investigated the potential link between Essential Tremor (ET) and longevity. They included 1,493 participants from 145 families, divided into ET-positive (ET+) and ET-negative (ET-) groups. The analysis aimed to clarify whether ET status is associated with lifespan. Results showed that the median age at death for deceased ET+ individuals was 80 years, compared to 70 years for ET- individuals. Living ET+ individuals had a median age of 63 years, while ET- individuals had a median age of 60 years. Survival analysis revealed a significantly longer lifespan for the ET+ group, with a hazard ratio of 0.44, indicating a lower mortality risk in ET+ patients, particularly among males.

I believe the study presents several methodological weaknesses that undermine the reliability and validity of its conclusions:

We are sincerely grateful for the time and effort you have dedicated to evaluating our work. In response to your comments, we have carefully reviewed and revised the manuscript once again, addressing your concerns to the best of our ability. Specifically, we have addressed issues related to potential confounding factors and study design limitations. Furthermore, we have included new content and reorganized key sections to enhance the overall narrative. Below, we provide a detailed, point-by-point response to your feedback. We hope these revisions comprehensively address your concerns and significantly strengthen the manuscript.

Specific comments:

- The study relies heavily on retrospective data for deceased patients, reconstructed through family trees. This approach introduces substantial recall bias, particularly regarding the accuracy of death diagnoses, age at death, and the presence of tremor. The reliance on family-reported data significantly undermines the validity and reliability of these key

Author Response: Study designs for analyzing aging and/or longevity in essential tremor cases include both retrospective and prospective models. Examples from the literature include:

1. Retrospective Design: One notable retrospective study is based on medical record review utilizing the records linkage system of the Rochester Epidemiology Project in the United States. This study encompassed individuals of all age groups, providing a comprehensive analysis of essential tremor cases across the lifespan (1).

2. Prospective Design: A prominent example of a prospective, population-based study is the NEDICES study in Spain, which focused on individuals aged 65 and older. This study employed a survey-based approach involving family members of elderly individuals. Death was systematically ascertained through data collection on age at death and cause of death from healthcare providers and family members (2).

3.Family-Based Longevity Study: Another approach is a family-based longevity study that evaluated relatives of patients with Parkinson’s disease, essential tremor, and control subjects from the same families. This study collected information on age at death and disease status using a similar survey-based approach with input from family members (3).

As noted in our previous response and discussed in the manuscript, we acknowledge that collecting death information from family members is a limitation of our study. Efforts to access original death certificates were hindered by complex bureaucratic processes and, in some cases, the reluctance of family members to engage in these procedures. We discussed this issue in the Limitations and Challenges part of the Discussion section.

Revised Text (Page 16, Lines 575–592):

Additionally, the lack of detailed cause-of-death information, particularly for older individuals or those from rural areas, limits our ability to explore specific mortality patterns in depth. Deaths categorized as “natural causes” often lack specificity, potentially obscuring critical insights into comorbidities and other factors influencing survival. Access to more precise cause-of-death data, such as death certificates or autopsy reports, would provide valuable insights into whether specific comorbidities or conditions, such as cardiovascular disease or neurodegenerative disorders, are common in ET patients compared to the general population. Furthermore, our exclusion of individuals who died from non-age-related causes (e.g., accidents or suicides) may introduce bias by narrowing the study population to those with age-related mortality. While this approach was necessary to focus on longevity, it may limit the generalizability of our findings. We cannot entirely rule out this possibility, however our analysis of individuals aged 60 and older still demonstrated a significant increase in survival times for ET patients (Log-rank p = 5.29 × 10-16 (Fig 3B)), suggesting that the longevity benefit associated with ET is robust even among older individuals. Additionally, our study did not account for certain lifestyle factors, such as diet, physical activity, and alcohol consumption, which may influence both ET development and longevity. Further studies that incorporate detailed data on these factors would help to clarify their potential impact.

That said, we emphasize the robustness of our data collection methods. Recruitment occurred during family appointments where multiple members were present, allowing for coordinated and cross-verified responses. Pedigree drawings and documentation of living and deceased first- and second-degree relatives, as well as tremor reports, were completed collaboratively and cross-checked in real time. These efforts were designed to enhance the specificity and accuracy of the data.

Revised Text (Page 5, Lines 144–152):

The accuracy of the pedigree information was critical for genetic analysis, and multiple checks were incorporated into the data collection process to ensure reliability and completeness. Recruitment of family members was coordinated during follow-up appointments, and efforts were made to have as many family members present as possible during these visits. This strategy facilitated the construction of robust pedigree charts, as family members could cross-verify information, minimizing discrepancies. During these appointments, family members collaborated to name and list all living and deceased first- and second-degree relatives, and they jointly reported on the presence of tremors in the family. This cross-referencing process ensured high specificity and accuracy in the familial data collected.

While a longitudinal prospective study would be ideal, it is not feasible given the retrospective nature of our research and the high proportion of deceased individuals. To address this limitation, we tested our data collection methods using 10 well-characterized multigenerational families, encompassing both affected and unaffected members. Despite the small sample size, we confirmed our observations in all families, with significant differences evident in 3 of the 10 families. Furthermore, we compared the survival status of ET patients with and without clinical assessments and found no significant difference in age at death between these groups (Figure 2E).

Revised Text (Page 16, Lines 565–575):

Despite its strengths, our study has limitations that warrant consideration. First, 35.6% of ET+ cases and 44.3% of ET- individuals were deceased family members, and 22 individuals had died after the baseline evaluation. For many of these individuals, assessments were based on family-reported histories rather than direct clinical evaluations, which introduces the potential for recall bias and misclassification of ET status. To mitigate this, we compared survival data between ET patients with and without clinical assessments, finding a significant difference in age at death between the two groups (U = 2.99 × 103, p = 2 × 10-3) (Fig 2E). Furthermore, we examined 10 phenotypically well-characterized multigenerational families, finding that in 3 of the 10 families, ET patients lived longer than their non-ET relatives. This finding suggests that genetic factors contributing to ET may also play a role in influencing longevity.

References:

1. Rajput AH, Offord KP, Beard CM, Kurland LT: Essential tremor in Rochester, Minnesota: a 45-year study. J Neurol Neurosurg Psychiatry 1984; 47: 466–470.

2. Benito-Leon J, Bermejo-Pareja F, Louis ED: Incidence of essential tremor in three elderly populations of central Spain. Neurology 2005; 64: 1721–1725.

3. Jankovic J, Beach J, Schwartz K, Contant C. Tremor and longevity in relatives of patients with Parkinson's disease, essential tremor, and control subjects. Neurology. 1995 Apr;45(4):645-8.

- The inclusion of participants from the same families raises concerns about genetic and environmental confounding. Family-based sampling does not control for hereditary factors that could influence both tremor and longevity. The diversity in socio-economic backgrounds within the sample is insufficient to ensure the external validity of the results, as it does not reflect the heterogeneity of the general population.

Author Response: We appreciate the reviewer’s concerns regarding genetic and environmental confounding and the external validity of our findings. As noted in the literature (1-4), family-based sampling is a powerful approach for jointly studying genetic and environmental factors, offering unique insights that may not be captured in population-based studies. While family-based cohorts inherently focus on hereditary aspects, they also allow for robust intra-family comparisons under shared environmental conditions (5-14). We believe this design is an advantage for investigating the genetic epidemiology of essential tremor (ET).

Our cohort predominantly consists of families residing in small villages, where members share similar socioeconomic conditions, dietary habits, and microbial exposures. This homogeneity helps mitigate potential confounding factors across different families. To further address potential biases, we conducted separate analyses within each family (Fig. 4) and compared these results to pooled analyses across all families (Fig. 1). Both approaches yielded consistent findings, supporting the validity of our conclusions.

However, we agree that the generalizability of findings from family-based cohorts to the broader population must be interpreted cautiously. To address this limitation, we have emphasized in the manuscript the importance of complementing family-based studies with population-based research. Our expanded discussion highlights the need for additional studies to explore the interplay between genetic and environmental factors in ET and its potential impact on longevity. Relevant revisions to the manuscript are as follows:

Revised Text (Page 3, Lines 73–103):

While neurodegenerative diseases are generally associated with shortened life expectancy, the impact of ET on longevity remains unclear [17–20]. Early observational studies such as that by Minor, suggested a potential survival advantage for individuals with ET [17,18], yet subsequent research has produced contradictory findings. For example, a longitudinal retrospective study conducted in 1984 by Rajput et al., on 266 ET patients, found no evidence of increased longevity. However, this study was limited by its relatively young cohort and insufficient follow-up into advanced age [19]. Conversely, a more comprehensive study by Jankovic et al. in 1995 examined 201 ET patients and compared them to 465 age-matched controls in a longitudinal prospective design. Their results indicated significantly increased longevity among ET patients with a relative risk [RR] of 1.59, (p = 0.01) [20]. More recently, a population-based prospective study conducted in three villages in central Spain in 2007 by Louis et al. suggested that ET may even increase mortality risk under certain conditions [21]. These conflicting results highlight the need for further investigation into the complex relationship between ET and lifespan.

To address these uncertainties further investigation into the relationship between ET and mortality is crucial. There is a clear need for more rigorous, family-based studies to investigate how ET might influence survival rates, particularly by accounting for genetic factors and controlling for confounding variables such as diet and socioeconomic status. Additionally, there is a lack of data exploring gender-specific differences in mortality and how such differences could contribute to our understanding of ET’s impact on longevity. In our comprehensive family-based retrospective study, we aimed to address this gap by comparing mortality risk in ET patients to their family members. By comparing survival outcomes among individuals with and without ET, and accounting for potential confounding factors, this work seeks to provide new insights into the potential link between ET and extended lifespan. In this study, we examined the pedigrees of 145 probands and tested the association of ET with longevity in 1,493 subjects. Our findings suggest that ET could indeed be associated with longer life expectancy, offering an intriguing avenue for further research into the underlying mechanisms that might confer this potential longevity advantage. These findings, if confirmed in large-scale longitudinal studies and biological investigations, could have significant implications for public health, particularly in understanding the complex interplay between ET, aging and survival.

Revised Text (Page 13-14, Lines 460–481):

Genetic Contributions to Longevity in ET. Genetic factors are likely to contribute significantly to the observed association between ET and increased lifespan. Several loci associated with ET, including those implicated in mitochondrial function and oxidative stress response, could influence both the development of tremor and the broader biological pathways governing aging. These genetic variations may provide protection against age-related diseases, such as cardiovascular conditions or metabolic syndromes, which are leading causes of mortality in the general population. Our family-based study design offers unique insights into the potential genetic basis of ET-related longevity. By comparing ET-positive (ET+) individuals to their ET-negative (ET-) relatives, we were able to control for many environmental and socioeconomic variables. By studying family members from the same households and communities, we sought to minimize these confounding variables. However, this design also introduces the potential for selection bias, as the families we studied may not represent the broader population. The genetic uniqueness of our cohort is underscored by an inbreeding coefficient of 36.5%, calculated from exome sequencing data, indicating a higher degree of genetic relatedness within these families than would be typical in the general population. The consistent trend of longer lifespans among ET+ individuals within the same families underscores the possibility that genetic factors influencing ET may also contribute to longevity. Our findings provide important insights into the relationship between ET and mortality. We found that the median age of death for the ET+ group was significantly higher than for ET- individuals (80 years vs. 70 years), suggesting that ET might be associated with increased longevity. This was further supported by survival analysis, which revealed a statistically significant higher risk of mortality in ET- individuals compared to those with ET.

References

1. Hopper, J. L., Bishop,

---

## [Decision Letter · Decision Letter 2]

19 Feb 2025

Effects of essential tremor on longevity and mortality rates in families

PONE-D-24-19093R2

Dear Dr. Onur Emre Onat,

We’re pleased to inform you that your manuscript has been judged scientifically suitable for publication and will be formally accepted for publication once it meets all outstanding technical requirements.

Kind regards,

Amina Nasri

Academic Editor

PLOS ONE

Reviewers' comments:

Reviewer's Responses to Questions

**Comments to the Author**

1. If the authors have adequately addressed your comments raised in a previous round of review and you feel that this manuscript is now acceptable for publication, you may indicate that here to bypass the “Comments to the Author” section, enter your conflict of interest statement in the “Confidential to Editor” section, and submit your "Accept" recommendation.

Reviewer #2: All comments have been addressed

2. Is the manuscript technically sound, and do the data support the conclusions?

Reviewer #2: Yes

3. Has the statistical analysis been performed appropriately and rigorously? 

Reviewer #2: Yes

4. Have the authors made all data underlying the findings in their manuscript fully available?

Reviewer #2: Yes

5. Is the manuscript presented in an intelligible fashion and written in standard English?

Reviewer #2: Yes

6. Review Comments to the Author

Reviewer #2: The paper analyzed data from 1,493 individuals across 145 families, encompassing both ET-positive (ET+) and ET-negative (ET-) participants.It is well structured but I suggest to revise the language.

7. PLOS authors have the option to publish the peer review history of their article (what does this mean? ). If published, this will include your full peer review and any attached files.

**Do you want your identity to be public for this peer review?** For information about this choice, including consent withdrawal, please see our Privacy Policy .

Reviewer #2: No

---

## [Editor Report · Acceptance letter]

PONE-D-24-19093R2

PLOS ONE

Dear Dr. Onat,

I'm pleased to inform you that your manuscript has been deemed suitable for publication in PLOS ONE. Congratulations! Your manuscript is now being handed over to our production team.

Kind regards,

on behalf of

Dr. Amina Nasri

Academic Editor

PLOS ONE